# Photosynthesis, Chlorophyll Fluorescence, and Hormone Regulation in Tomato Exposed to Mechanical Wounding

**DOI:** 10.3390/plants13182594

**Published:** 2024-09-17

**Authors:** Hui Yan, Kai Fu, Jiajia Li, Mingyong Li, Shaofan Li, Zhiguang Dai, Xin Jin

**Affiliations:** College of Agricultural Equipment Engineering, Henan University of Science and Technology, Luoyang 471000, China; hnyanhui@yeah.net (H.Y.); 18203173207@163.com (K.F.); 15237726147@163.com (J.L.); mingyonglee@163.com (M.L.); 18568332369@163.com (S.L.); daizhiguang100@163.com (Z.D.)

**Keywords:** photosynthetic limiting factors, PSII efficiency, hormonal crosstalk, principle component analysis, mechanical wounding

## Abstract

To understand the physiological responses of seedlings to mechanical wounding, we analyzed photosynthesis, chlorophyll fluorescence, and endogenous hormones in tomato (*Solanum lycopersicum* L.) subjected to varying levels of mechanical pressure. The results showed that, at 4 h after wounding, excess excitation energy was dissipated as thermal energy through the reduction in the photosystem II (PSII) opening degree and the increase in non-photochemical quenching. Photodamage was avoided, and stomatal closure was the most prominent factor in photosynthesis inhibition. However, 12 h after wounding, the photoprotective mechanism was insufficient to mitigate the excess excitation energy caused by the wound, leading to photochemical damage to physiological processes. Meanwhile, the non-stomatal factor became the most prominent limiting factor for photosynthesis at 80 N pressure. At 12 and 36 h after wounding, the concentrations of abscisic acid (ABA), methyl jasmonate (MeJA), indole-3-acetic acid (IAA), zeatin riboside (ZR), and gibberellic acid (GA_3_) in the stems showed a trend towards being increased, which promoted wound healing. However, after mechanical wounding, the ratio of stress- to growth-promoting hormones first increased and then decreased. This pattern can enhance stress resistance and promote cell division, respectively. Comprehensive analysis showed that the fluorescence parameter, photochemical quenching coefficient (*Qp*_Lss), was the most suitable indicator for evaluating mechanical wounding conditions.

## 1. Introduction

Currently, automatic transplanting devices are widely used in seedling transplanting operations. These devices not only improve seedling survival rates and transplant quality but also reduce labor intensity and planting costs. However, owing to various factors, such as individual seedling differences and changes in clamping force, the negative effects generated by transplanting operations on seedling quality are easily caused during automatic transplanting. For instance, during the seedling-picking process, stem damage is frequently caused by the excessive clamping force of the transplanting devices [1]. Currently, preventing stem-clamping damage during the transplanting process has become a key issue that needs to be addressed. Hence, many efforts have been made in recent years to reduce the mechanical wounding caused by stem-clamping devices. However, these efforts had limited success, primarily because of insufficient information regarding the effect of stem clamping on the physiological characteristics of seedlings.

Several studies have reported that mechanical wounding is a common stress factor that causes changes in gene expression and signal transduction [2,3]. As a result, various physiological characteristics, including hormone levels, stomatal aperture, transpiration, and photosynthesis, are affected. First, many endogenous signaling molecules are induced by mechanical wounding that are involved in wound healing and plant growth. These include abscisic acid (ABA), methyl jasmonate (MeJA), indole-3-acetic acid (IAA), zeatin riboside (ZR), and gibberellic acid (GA_3_), which have the most important functions [4,5]. As plant stress hormones, ABA and MeJA are synthesized in large quantities under mechanical wounding and play critical roles in signaling and healing processes [6,7,8]. However, hormonal interactions, rather than individual signaling, may regulate the wound-healing process. Plant growth-promoting hormones, such as IAA, ZR, and GA_3_, are also involved in regulating wound-healing processes [9]. Although the involvement and changes in individual hormones caused by mechanical wounding have been determined in many studies [10,11], to date, little information has been provided regarding the crosstalk and balance between stress and growth-promoting hormones.

Hormonal signals may be involved in the regulation of several physiological processes. The stomatal aperture, which plays a vital role in controlling CO_2_ and H_2_O exchange, is reduced by mechanical wounding owing to hormonal crosstalk between ABA and ZR [12]. Partial stomatal closure caused by wounding primarily restricts external CO_2_ diffusion into the mesophyll cells. When intercellular CO_2_ concentrations are insufficient for meeting the demand for photosynthetic carbon assimilation, stomatal factors are involved in restricting photosynthesis [13]. Photosynthetic inhibition caused by stomatal closure may trigger an imbalance between electron supply and photosynthetic activity, subsequently leading to the over-excitation of absorbed energy [14]. With the downregulation of PSII efficiency and the increase in thermal dissipation, excess excitation energy could be dissipated, potentially preventing damage to PSII reaction centers. However, under severe damage, the photoprotective mechanism may be insufficient for preventing PSII damage, and non-stomatal factors may be involved in limiting photosynthesis [15]. Normally, stomatal and non-stomatal factors limit photosynthesis. However, the major factor responsible for photosynthesis inhibition remains controversial. Mechanical wounding can cause changes in endogenous hormones, photosynthesis, and fluorescence; therefore, mechanical wounding conditions are closely related to these parameters. To determine the extent of stem wounds during transplanting operations, it is necessary to analyze the physiological processes involving hormones, photosynthesis, and fluorescence in response to transplanting operations. In addition, it is important to identify key physiological parameters that reflect wounding conditions using principal component analysis (PCA). Therefore, the physiological changes caused by the transplanting devices should be assessed.

Tomato (*Solanum lycopersicum* L.), one of the most important horticultural crops in the world, has high nutritive value because it is rich in vitamins A, B, C, and carotene, as well as inorganic salts, including potassium, magnesium, calcium, iron, zinc, copper, iodine, and phosphorus [16]. However, during tomato seedling-transplanting operations, the use of stem-clamping devices may lead to mechanical damage to the stem, thereby decreasing the transplanting quality and ultimately reducing the tomato yield and fruit quality. To understand the physiological response of tomato to mechanical damage, we studied photosynthesis, chlorophyll fluorescence, hormonal changes, and stem balance under different levels of mechanical pressure. A schematic diagram of the experimental design is shown in Figure 1. The aim of the experiment was (1) to investigate the impact of mechanical damage on photosynthetic gas exchange and excitation energy utilization and dissipation in PSII; (2) to investigate the variations in hormonal levels and balance in the stems of tomato under mechanical wounding conditions and the role of hormones in coordinating physiological functions; (3) to screen the key parameters reflecting the mechanical wounding conditions from various physiological parameters.

## 2. Materials and Methods

### 2.1. Plant Cultivation

The study was conducted at the Henan University of Science and Technology. Seeds of tomato (*S. lycopersicum* L. cv Jinhe 40) were sown into frustum-shaped nutritional bowls (upper length 3.5 cm, lower length 1.5 cm, and height 3.5 cm) filled with peat-based substrate in a sunlit greenhouse. Four-week-old seedlings were transferred to a growth chamber (200 μmol m^−2^ s^−1^ photosynthetic photon flux density under a 12 h photoperiod from 8:00 a.m. to 20:00 p.m., a relative humidity of 70%, and a day/night temperature of 25/20 °C) for five days of physiological acclimatization before treatment.

### 2.2. Compression Treatment

To imitate the mechanical wounding caused by stem clamping, compression treatments were performed at 5:00 a.m. using a DR-506 strength tensile testing machine (Dongguan Dongri Instrument Co., Ltd., Dongguan, China) with a pair of clamped disks (diameter, 108.65 mm). After physiological acclimatization, the tomato seedlings were placed horizontally on the lower disk of the tensile-strength testing machine, and three different compression pressures (0, 40, and 80 N) were applied to the stems of the seedlings. The compression speed of the upper plate was 20 mm s^−1^ and the compression pressure on the stem of the seedlings lasted for three seconds.

### 2.3. Photosynthesis Measurements

The second fully expanded functional leaves were selected for the photosynthetic measurements. At 4, 12, and 36 h after mechanical wounding, the photosynthetic rate (*P*_n_), stomatal conductance (*g*_s_), transpiration rate (T_r_), and intercellular CO_2_ concentration (*C*_i_) were determined using a portable photosynthesis system (LI-6400; LI-COR Biosciences, Inc., Lincoln, NE, USA) with a CO_2_ concentration of 390 cm^3^ m^−3^ and a light intensity of 1000 μmol photon m^−2^ s^−1^.

### 2.4. Fluorescence Measurements

The leaves selected for photosynthetic measurements were used for chlorophyll fluorescence measurements. The minimum fluorescence (*F_o_*), maximum fluorescence (*F_m_*), variable fluorescence (*F_v_*), potential PSII activity (*F_v_/F_o_*), maximal photochemical efficiency (*F_v_/F_m_*), actual photochemical efficiency (*Qy_L_ss_*), photochemical quenching coefficient (*QP_L_ss_*), and steady-state non-photochemical quenching (*NPQ_L_ss_*) were determined using a handheld chlorophyll fluorometer (FluorPen FP 110; Photon Systems Instruments Ltd., Drásov, Czech Republic), as described by Prasad [17].

### 2.5. Phytohormone Assay

After performing photosynthesis and fluorescence measurements, the stems were harvested and ground immediately into a fine powder in liquid nitrogen. The concentrations of ABA, IAA, ZR, GA_3_, and MeJA in the stems were determined using enzyme-linked immunosorbent assay (ELISA), as described by He [18].

### 2.6. Statistical Analysis

Data were analyzed using SPSS v16.0 (SPSS, Inc., Chicago, IL, USA). The treatment means were compared using Tukey’s HSD test to identify significant differences at a significance level of *p* = 0.05. The PCA was performed using MetaboAnalyst 5.0.

## 3. Results

### 3.1. Photosynthetic Gas Exchange

The evolution of photosynthetic gas exchange in tomato after mechanical wounding is shown in Figure 2. The non-wounded seedlings maintained a photosynthetic rate (*P*_n_) of approximately 15 μmol CO_2_ m^−2^ s^−1^, a stomatal conductance (*g*_s_) of approximately 0.155 mol H_2_O m^−2^ s^−1^, and a transpiration rate of around 5.5 mmol H_2_O m^−2^ s^−1^. Compared with the non-wounded seedlings, *P*_n_, *g*_s_, and T_r_ showed a trend towards reduction after 4 and 12 h of mechanical wounding. However, after 36 h of mechanical wounding, the *g*_s_ and T_r_ were restored to the levels of non-wounded seedlings. The changes in *C*_i_ were different from those in the photosynthetic parameters. They showed a trend towards reduction after 4 or 36 h of stress, but they significantly increased after 12 h of 80 N stress.

### 3.2. Chlorophyll Fluorescence

The chlorophyll fluorescence in tomato after mechanical wounding was analyzed in our study (Figure 3 and Figure 4). At 4 and 36 h after mechanical wounding, there were no significant differences between wounded and non-wounded seedlings in the minimum fluorescence (*F*_o_), maximum fluorescence (*F*_m_), variable fluorescence (*F*_v_), maximum efficiency of PSII (*F*_v_/*F*_m_), as well as the potential efficiency of PSII (*F*_v_/*F*_o_). However, after 12 h of 80 N stress, *F*_o_ increased significantly, but *F*_m_, *F*_v_, *F*_v_/*F*_m_, and *F*_v_/*F*_o_ were significantly reduced compared with those of the non-wounded seedlings. The changes in the photochemical quenching coefficient (*qP*_Lss) and steady-state non-photochemical quenching (*NPQ*_Lss) differed from those in the other chlorophyll parameters. After 4 h of wounding stress, as well as 12 and 36 h of 80 N stress, *qP*_Lss decreased significantly, whereas *NPQ*_Lss gradually increased.

### 3.3. Concentrations of Stress-Responsive and Growth-Promoting Hormones in Stems

The stress-responsive and growth-promoting hormones play vital roles in regulating plant physiology; therefore, the variations in ABA, MeJA, IAA, ZR, and GA_3_ concentrations in the stems of *S. lycopersicum* were analyzed (Figure 5). Compared with the non-wounded seedlings, the concentrations of ABA, MeJA, IAA, and GA_3_ in the stems significantly increased 4 h after wounding. At 12 and 36 h after wounding, the concentrations of stress-responsive and growth-promoting hormones in the stems showed a trend towards increase compared with those in the non-wounded seedlings.

### 3.4. Ratios of Stress-Responsive Hormones to Growth-Promoting Hormones in the Stems

To assess the relative changes in stress-responsive hormones and growth-promoting hormones, the ratios of ABA or MeJA to IAA, ZR, and GA_3_ in the stems were analyzed in our study (Figure 6). At 4 h after wounding, the levels of ABA/IAA and MeJA/IAA in the stems were significantly reduced, whereas ABA/ZR, MeJA/ZR, and ABA/GA_3_ showed a trend toward increase. Compared with non-wounded seedlings, the levels of ABA (MeJA)/IAA (ZR or GA_3_) in the stems showed a trend toward increase at 12 h after wounding but exhibited a decreasing trend after 36 h of 80 N stress.

### 3.5. PCA of Physiological Parameters

To evaluate the responses of *S. lycopersicum* to mechanical wounding at different times after treatment, a PCA was performed based on the parameters, including photosynthesis, chlorophyll fluorescence, plant hormones, and hormone balance. We extracted the top four principal components (PC1 to PC4) in our analysis, which collectively accounted for approximately 85% of all variations. Among the 22 physiological parameters, ABA, *Qp*_Lss, *C*_i_, and ZR were the primary factors for each principal component (Table 1). In addition, the physiological responses of *S. lycopersicum* to mechanical wounding were distinct at various time points after the treatment. Compared with 4 and 36 h after wounding, mechanical wounding after 12 h had the most significant effects on the physiological parameters of *S. lycopersicum* (Figure 7).

## 4. Discussion

Respiration and photosynthesis are the most important physiological processes by which plants obtain and utilize resources and energy [19]. Numerous studies have reported that mechanical wounding, which is an abiotic stress factor, can decrease respiration and photosynthesis [3,20]. The reduction in respiration may be due to mechanical wounding accelerating ABA synthesis and inducing stomatal closure. The reduction in photosynthesis can be attributed to various factors, such as stomatal closure, repression of photosynthetic genes, and chlorophyll decomposition [21]. Thus, in response to mechanical wounding, both stomatal and non-stomatal factors are involved in photosynthesis inhibition. Stomatal closure inhibits CO_2_ influx and decreases intercellular CO_2_ levels. However, the non-stomatal limitations of photosynthesis reduce intercellular CO_2_ consumption and accelerate CO_2_ accumulation [22]. Therefore, the prominent factor that inhibits photosynthesis in response to mechanical wounding can be determined by variations in the intercellular CO_2_ levels [15]. During our experiment, reductions in *P*_n_ and *g*_s_ were demonstrated, along with a decrease in *C*_i_ at 4 h after mechanical wounding. This suggests that stomatal closure caused by mechanical wounding was the most prominent factor inhibiting photosynthetic carbon assimilation. However, reductions in *P*_n_ and *g*_s_ were accompanied by an increase in *C*_i_ at 12 h after applying 80 N pressure, indicating that non-stomatal factors were the most prominent factors inhibiting photosynthesis under severe wounding. After 36 h of mechanical wounding, there were no significant differences in *P*_n_ and *g*_s_ between the wounded and non-wounded seedlings, suggesting that the effect of mechanical wounding was partially restored.

Chlorophyll fluorescence analysis, one of the most widely used and powerful techniques, is not only used for evaluating photosynthetic performance but also serves as a reliable indicator of plant stress responses [23]. In our study, we used this technique to investigate the changes in fluorescence parameters in response to mechanical wounding. Compared with the non-wounded seedlings, no significant differences in *F*_o_, *F*_m_, and *F*_v_ were detected at 4 h after mechanical wounding, indicating that photochemical damage to the PSII reaction center might not have occurred. However, after 12 h of 80 N stress, the *F*_o_ value significantly increased compared with that of the non-wounded seedlings. This indicated that the light-harvesting complex II (LHCII) might be dissociated from the PSII reaction centers in response to mechanical wounding, leading to a reduction in the efficiency of excitation energy transfer from the chlorophyll molecules of LHCII to the PSII reaction centers [24]. Additionally, significant decreases in *F*_m_ and *F*_v_ were observed at 12 h after mechanical wounding. This could be attributed to the incomplete reduction in electron acceptors in the electron transfer chains. In addition, the photochemical activity of PSII decreased because of mechanical wounding [24,25]. At 36 h after mechanical wounding, the values of *F*_o_, *F*_m_, and *F*_v_ were restored compared with those of the non-wounded seedlings, suggesting that photochemical damage to PSII was alleviated.

At 4 h after mechanical wounding, *qP*_Lss significantly decreased, whereas *NPQ*_Lss increased significantly compared with the non-wounded seedlings. *qP*_Lss and *NPQ*_Lss reflect the opening degree of PSII centers and energy dissipation in the form of thermal energy [26]. The decrease in *qP*_Lss and increase in *NPQ*_Lss is a photoprotective mechanism in which the opening degree of PSII centers is reduced and more excitation energy is dissipated as thermal energy rather than being involved in driving photosynthesis [27]. In this way, photochemical damage to PSII centers triggered by excess excitation energy was avoided and *F*_v_/*F*_m_ and *F*_v_/*F*_o_ were maintained. However, 12 h after mechanical wounding, with a decrease in *qP*_Lss and an increase in *NPQ*_Lss, *F*_v_/*F*_m_, and *F*_v_/*F*_o_ showed decreasing trends. This suggests that the photoprotective mechanism is not adequate for alleviating wound-induced excess excitation energy and that photochemical damage is involved in physiological processes [28]. However, *F*_v_/*F*_m_, *F*_v_/*F*_o_, *qP*_Lss, and *NPQ*_Lss were restored to non-wounded levels at 36 h after mechanical wounding, indicating that photochemical damage was alleviated.

In addition to photosynthesis and fluorescence, mechanical wounding stimulates a series of plant secondary metabolism that are critical for wound healing in physiologically damaged seedlings [8]. Numerous studies have reported that phytohormones are involved in both the abiotic and biotic stress responses [29,30]. In this study, compared with those of the non-wounded seedlings, both ABA and MeJA in tomato stems increased after mechanical wounding (Figure 5). This phenomenon could be explained by the fact that ABA and MeJA biosynthesis is accelerated by wounding [6,31]. ABA and MeJA are crucial components of the wound signal transduction pathway in seedlings, and their accumulation due to wounding is important for wound healing, including the formation of closing layers and wound periderm formation [6]. Similarly, the concentrations of IAA, ZR, and GA_3_ in tomato stems increased after mechanical wounding (Figure 5). However, the reasons for the increased levels of the different hormones may differ. The increase in IAA may be due to both hydrolysis of IAA-ASP conjugates and accelerated biosynthesis of IAA. The increase in ZR may indicate differential tissue/cellular compartmentation, post-transcriptional regulation of enzyme activity, and so on [9]. Meanwhile, the increase in GA could be explained by the fact that, wounding induced the expression of GA biosynthesis-related genes, and GA biosynthesis is upregulated thereafter [32]. These are effective defense mechanisms and play vital roles in coordinating meristem activity, promoting wound periderm development, and reducing the negative effects induced by wounding [9,32].

Although the accumulation of ABA, MeJA, IAA, ZR, and GA_3_ is conducive to wound healing, excessive endogenous hormones are also involved in the regulation of plant growth. In addition, multiple hormones coordinate plant growth through antagonistic, synergistic, or additive effects [33]. ABA and MeJA are plant stress hormones that play vital roles in improving stress resistance and inhibiting cell division and elongation [34,35]. IAA, ZR, and GA_3_ are growth-promoting hormones that promote cell division and elongation. Therefore, ABA (MeJA)–IAA (ZR or GA_3_) exhibits complex hormonal crosstalk in growth regulation after mechanical wounding. In our study, the levels of ABA (MeJA)/IAA, ABA (MeJA)/ZR, and ABA (MeJA)/GA_3_ in the stems showed a trend toward increase at 12 h after wounding. This indicated that the stems suffered severe damage, leading to enhanced stress resistance and inhibited cell division and elongation. However, after 36 h of 80 N stress, ABA (MeJA)/IAA (ZR or GA_3_) showed a decreasing trend, indicating that stem damage was alleviated and cell division and elongation were promoted.

Mechanical wounding can cause a series of physiological changes, and a single physiological parameter may not be able to evaluate the extent of the wound effectively and comprehensively. Therefore, PCA, which simplifies the original complex variables to a small set of variables known as principal components through statistical transformations, was used to analyze and evaluate the vital physiological responses to mechanical wounding [36]. Our study revealed that, based on the PCA, four principal components (PC1 to PC4) accounted for approximately 85% of all variations. In the 22 physiological parameters, ABA, *Qp*_Lss, *C*_i_, and ZR were critical for each principal component, indicating that these hormonal, fluorescent, and photosynthetic indicators were sensitive to mechanical wounding. Conversely, during the automatic transplanting operations, hormonal, fluorescent, and photosynthetic indicators, such as ABA, *Qp*_Lss, *C*_i_, and ZR, should be monitored to evaluate the mechanical wounding conditions after stem clamping. Fluorescence analysis is a powerful technique for understanding how seedlings respond to environmental changes. It offers several advantages, such as being rapid and non-invasive [23,37]. Therefore, *Qp*_Lss is the most suitable indicator for evaluating mechanical wounding conditions. In addition, the results of the PCA also suggested that 12 h after mechanical wounding resulted in different physiological responses compared with 4 or 36 h after mechanical wounding. The effects of 12 h after wounding on the physiological responses of *S. lycopersicum* were more significant than those of 4 and 36 h after wounding. This indicates that physiological damage did not reach a peak at 4 h after wounding but was partially restored at 36 h after wounding.

## 5. Conclusions

In conclusion, at 4 h after wounding, the photoprotective mechanism prevented photodamage and stomatal closure became the prominent factor in limiting photosynthesis. However, photosynthetic non-stomatal limitation caused by photodamage became the prominent factor limiting photosynthesis at 12 h after wounding. During mechanical wounding, various endogenous hormones were increased to accelerate wound healing. However, the ratio of stress to growth-promoting hormones initially increased and then decreased to enhance stress resistance or promote cell division, respectively. *Qp*_Lss was the most suitable indicator for evaluating wound conditions.

## Figures and Tables

**Figure 1 plants-13-02594-f001:**
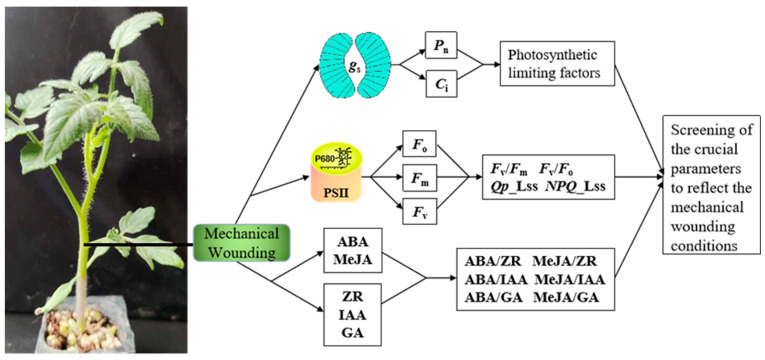
The schematic diagram of the experimental design.

**Figure 2 plants-13-02594-f002:**
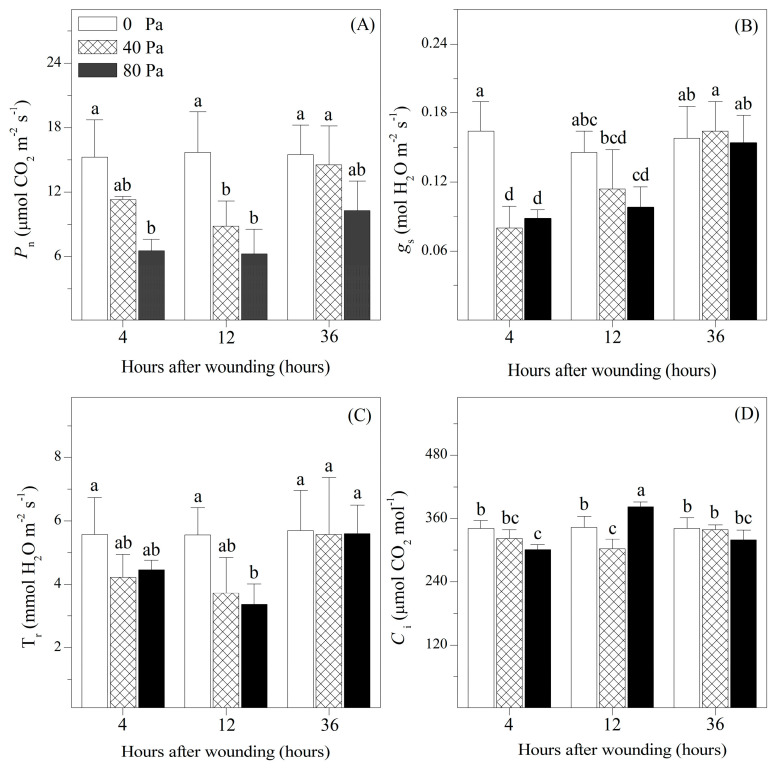
Changes in (**A**) photosynthetic rate (*P*_n_), (**B**) stomatal conductance (*g*_s_), (**C**) transpiration rates (T_r_), and (**D**) internal CO_2_ concentrations (*C*_i_) of *Solanum lycopersicum* L. under non-wounding and wounding conditions. Values are averages of 5 replicates ± s.e. Different letters refer to significant differences among treatments.

**Figure 3 plants-13-02594-f003:**
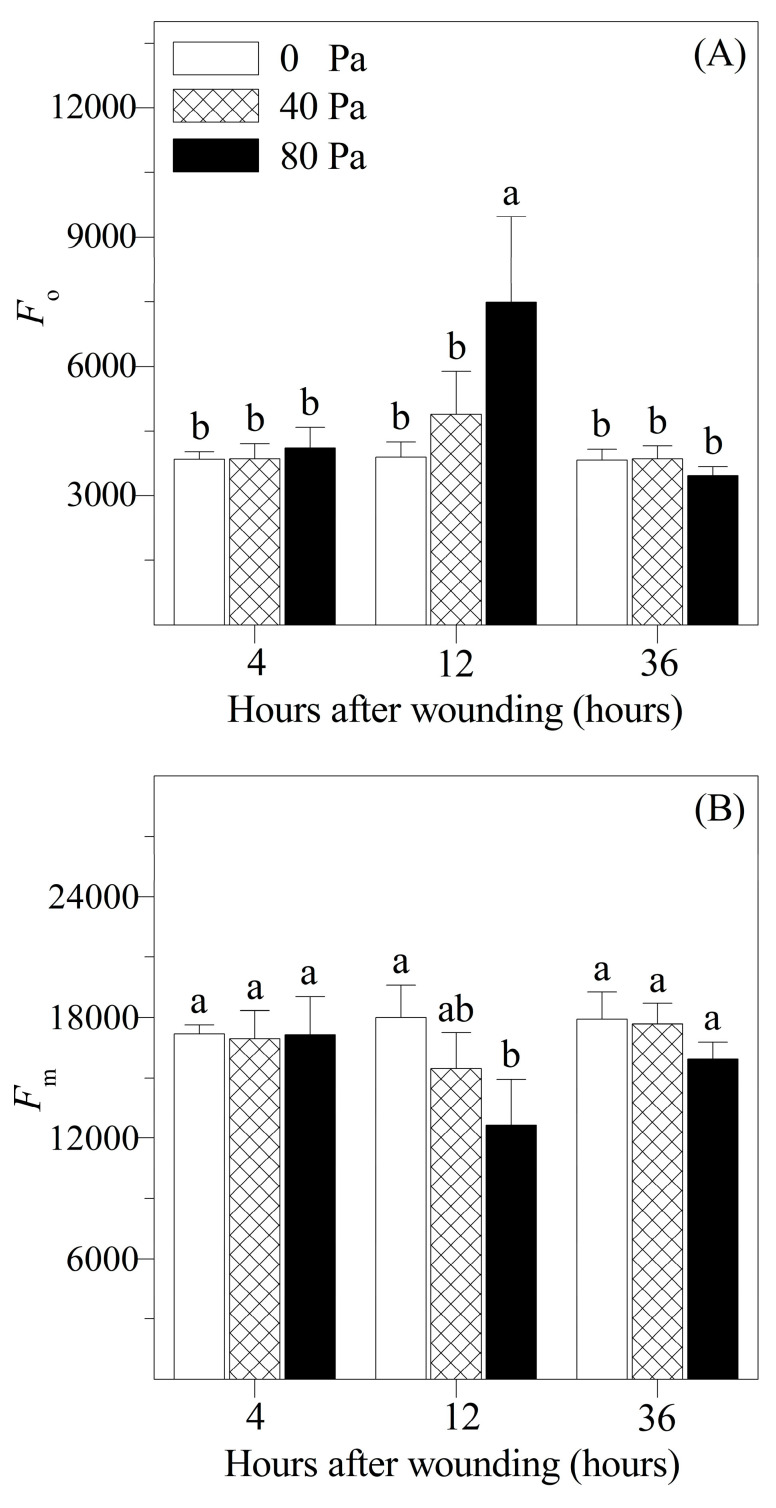
Changes in (**A**) minimum fluorescence (*F_o_*), (**B**) maximum fluorescence (*F_m_*), and (**C**) variable fluorescence (*F_v_*) of *Solanum lycopersicum* L. under non-wounding and wounding conditions. Values are averages of 5 replicates ± s.e. Different letters refer to significant differences among treatments.

**Figure 4 plants-13-02594-f004:**
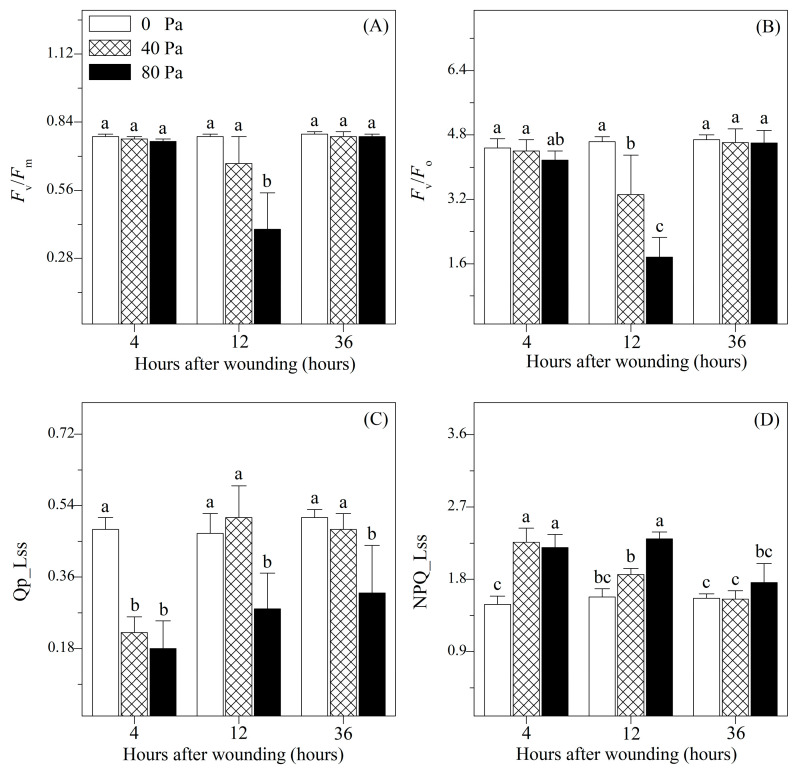
Changes in (**A**) maximal photochemical efficiency (*F_v_/F_m_*), (**B**) potential activity of PSII (*F_v_/F_o_*), (**C**) photochemical quenching coefficient (*QP_L_ss_*), and (**D**) steady-state non-photochemical quenching (*NPQ_L_ss_*) of *Solanum lycopersicum* L. under non-wounding and wounding conditions. Values are averages of 5 replicates ± s.e. Different letters refer to significant differences among treatments.

**Figure 5 plants-13-02594-f005:**
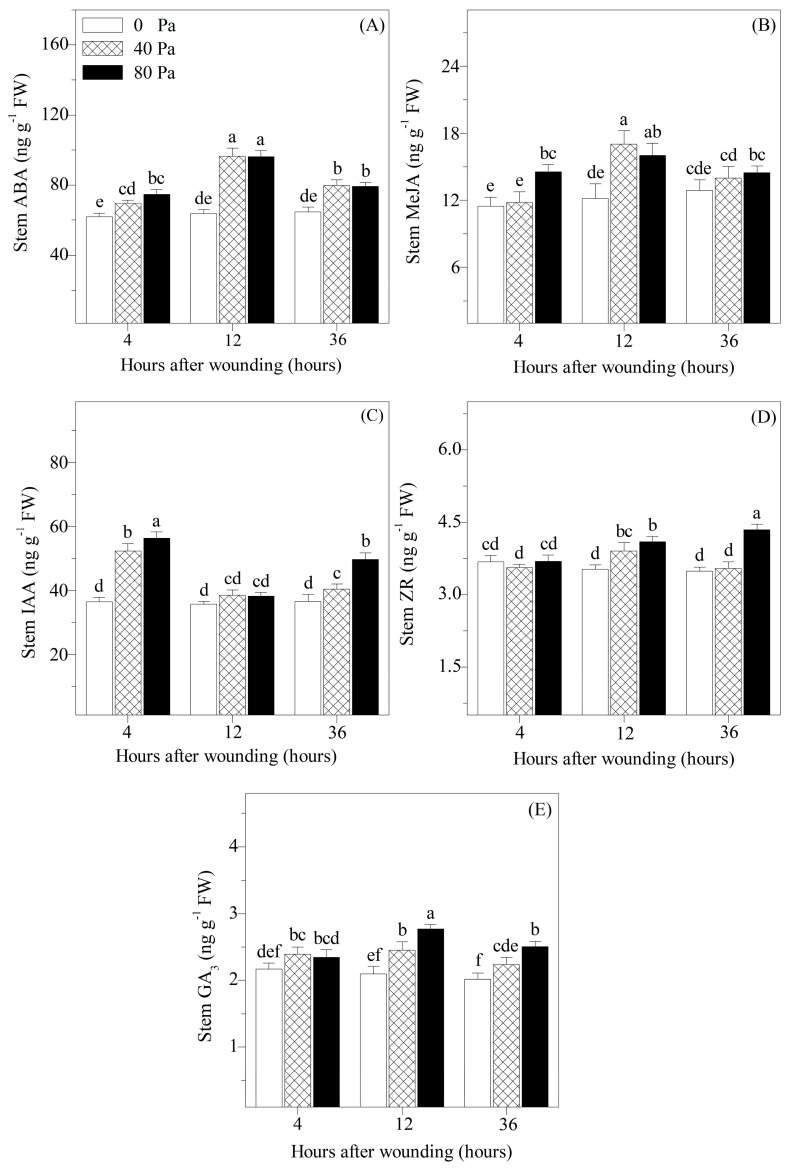
Changes in (**A**) stem abscisic acid (ABA), (**B**) stem methyl jasmonate (MeJA), (**C**) stem indole-3-acetic acid (IAA), (**D**) stem zeatin-riboside (ZR), and (**E**) stem gibberellic acid (GA_3_) of *Solanum lycopersicum* L. under non-wounding and wounding conditions. Values are averages of 5 replicates ± s.e. Different letters refer to significant differences among treatments.

**Figure 6 plants-13-02594-f006:**
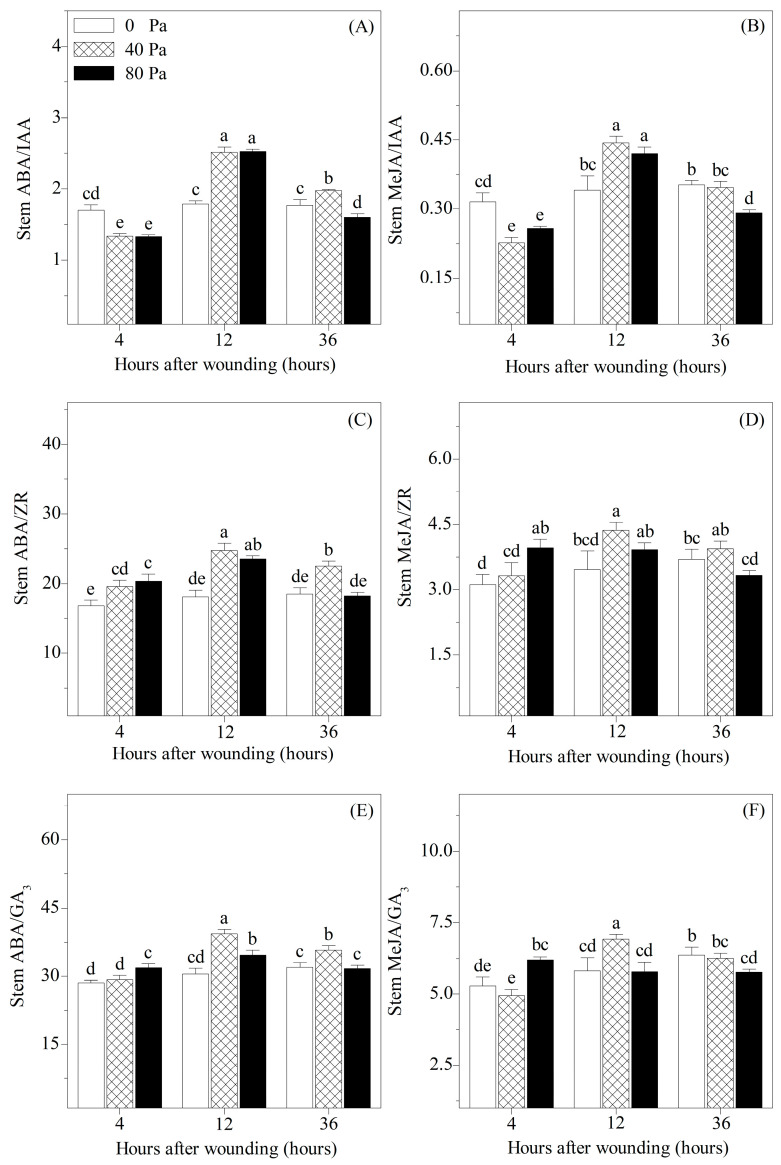
Changes in (**A**) stem ABA/IAA, (**B**) stem MeJA/IAA, (**C**) stem ABA/ZR, (**D**) stem MeJA/ZR, (**E**) stem ABA/GA_3_, and (**F**) stem MeJA/GA_3_, stem ABA/GA_3_, and stem MeJA/GA_3_ of *Solanum lycopersicum* L. under non-wounding and wounding conditions. Values are averages of 5 replicates ± s.e. Different letters refer to significant differences among treatments.

**Figure 7 plants-13-02594-f007:**
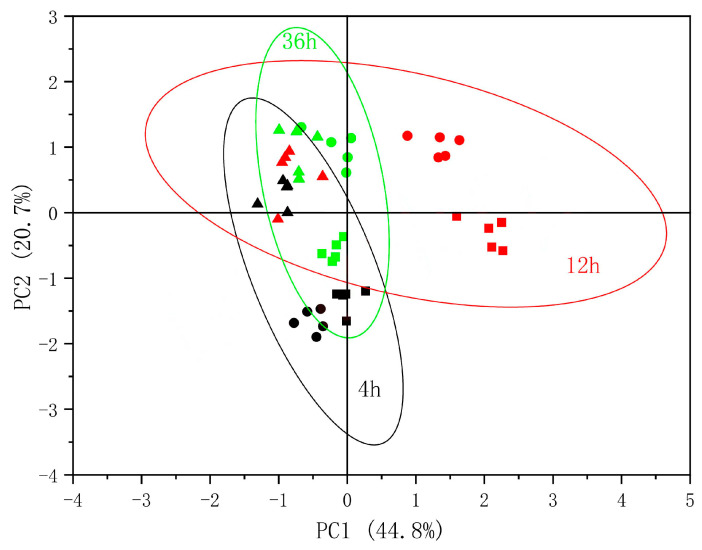
Principal component analysis (PCA) of physiological parameters in *Solanum lycopersicum* L. at 4 h (black), 12 h (red) and 36 h (green) after 0 N (triangle), 40 N (circle), and 80 N (square) mechanical pressures of wounding.

**Table 1 plants-13-02594-t001:** Principal component analysis (PCA) of physiological parameters in *Solanum lycopersicum* L. at 4 h, 12 h and 36 h after mechanical wounding.

Variables	PC1	PC2	PC3	PC4
*P* _n_	−0.20	0.22	−0.17	−0.02
T_r_	−0.20	0.15	−0.08	0.35
*g* _s_	−0.14	0.30	−0.14	0.39
*C* _i_	0.06	0.10	−0.39	−0.06
*F* _o_	0.24	−0.01	−0.28	−0.21
*F* _m_	−0.24	0.07	0.19	−0.24
*F* _v_	−0.27	0.05	0.26	−0.07
*F*_v_/*F*_m_	−0.26	0.03	0.29	0.09
*F*_v_/*F*_o_	−0.28	0.02	0.23	0.08
*Qp*_Lss	−0.07	0.41	0.00	−0.02
*NPQ*_Lss	0.17	−0.33	0.02	−0.20
ABA	0.30	0.03	0.12	0.14
MeJA	0.27	0.06	0.24	0.21
IAA	−0.01	−0.38	0.27	0.17
ZR	0.16	−0.09	−0.04	0.60
GA_3_	0.26	−0.19	−0.03	0.26
ABA/IAA	0.24	0.27	−0.09	−0.05
ABA/ZR	0.27	0.09	0.19	−0.16
ABA/GA_3_	0.22	0.23	0.24	−0.03
MeJA/IAA	0.21	0.33	−0.04	−0.03
MeJA/ZR	0.21	0.14	0.32	−0.12
MeJA/GA_3_	0.12	0.28	0.36	0.02
Proportion of variation (%)	44.77	20.73	13.35	6.27
Cumulative proportions of variation (%)	44.77	65.50	78.85	85.12

## Data Availability

Data are contained within the article.

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
