# Peer review of "Photosynthesis, Chlorophyll Fluorescence, and Hormone Regulation in Tomato Exposed to Mechanical Wounding"

_plants, 2024, doi:10.3390/plants13182594_

Round 1

Reviewer 1 Report

Comments and Suggestions for Authors

Just  few comments are inserted in text attached.

Your interesting paper is very clear and understanable. It is easy to read.

My only doubt is  to move or not Fig.6, and Fig. 7 from Discussion to the Results session.  I will leave the decision to authors.

Author Response

Response to Reviewer 1 Comments

Dear Professor

Thank you very much for your help. We are grateful for the constructive comments. We have addressed all the comments raised in the revised manuscript. Below is a detailed response to your comments.

Comments 1: Please, try to avoid the same words/phrases as in title

Response 1: Thank you very much. The keywords were revised to “Photosynthetic limiting factors  PSII efficiency  Hormonal crosstalk  Principle component analysis  Mechanical wounding”

Comments 2: Fig. 4 should be Fig. 5 in 3.3

Response 2: Thank you very much. The mistake was revised.

Comments 3: Fig. 5 should be Fig. 6 in 3.4

Response 3: Thank you very much. The mistake was revised.

Comments 4: Fig. 5 should be Fig.7 in 3.5

Response 4: Thank you very much. The mistake was revised.

Reviewer 2 Report

Comments and Suggestions for Authors

Dear authors,

In the manuscript titled with "Photosynthesis, chlorophyll fluorescence, and hormone regulation in tomato exposed to mechanical wounding", the authors observed variations of physiological parameters at different time points under different levels of mechanical wounding.

Author Response

Response to Reviewer 2 Comments

Dear Professor

Thank you very much for your help. We are grateful for the constructive comments. We have addressed all the comments. Below is a detailed response to your comments.

Comments 1: Authors described photosynthesis was measured between 9 and 11 am, but all results indicated 4, 12 and 36 hrs after wounding. A reviewer can not understand how this time point was possible to measure all physiological parameters. For instance, if 4 hrs is morning, 12 and 36 hrs are night. In this experimental condition, hormone levels as well as photosynthesis and fluorescence can entirely affected by a diurnal fluctuation. How does authors maintain CO2 concentration and light intensity?

Response 1: Thank you very much for helping us detect the experimental process which was not clearly described in our manuscript. The experimental process is as follows: Before experiment, the seedlings were cultivated in growth chamber (200 μmol m–2 s–1 photosynthetic photon flux density under a 12-hour photoperiod from 8:00 am to 20:00 pm). At 5:00 am, the compression treatments were performed and seedlings were wounded. At 9:00 am (4 hrs after wounding), 17:00 pm (12 hrs after wounding), and 17:00 pm (next day, 36 hrs after wounding), photosynthesis, fluorescence and other parameters were measured. During photosynthesis measurement, CO2 concentration and light intensity in leaf chamber of portable photosynthesis system were separately set to 390 cm3 m-3 and 1000 μmol photon m-2 s-1. The experimental process was not described clearly due to our negligence, we revised it in the revised manuscript. Thank you again!  

Comments 2: A reviewer does not agree with authors argument. Show an evidence of stomatal closure and cell division. What does ‘stress resistance’ mean? This description of conclusion makes a big jump in logic assessment.

Response 2: Thank you very much. The conclusion is ambiguous due to improper description. The conclusion was revised in the manuscript.

In the revised manuscript, the conclusion consists of five sentences.

The first and second sentences (In conclusion, ……at 12 hours after wounding) described the impact of mechanical wounding on photosynthesis and fluorescence. At 4 hours after wounding, stomatal closure became the prominent factor limiting photosynthesis. However, at 12 hours after wounding, the non-stomatal factor became the prominent factor limiting photosynthesis (Confirmed in the first paragraph of Discussion). At 4 hours after wounding, the photoprotective mechanism prevented photodamage. However, at 12 hours after wounding, photodamage occurred (Confirmed in the second paragraph of Discussion).

The third and forth sentences (During mechanical wounding, various endogenous……promote cell division, respectively.) described the changes of hormones and hormonal interactions after mechanical wounding. Among them, various endogenous hormones were increased, which would accelerate wound healing (Confirmed in the fourth paragraph of Discussion). And also, the ratio of stress to growth-promoting hormones initially increased and then decreased, which initially enhance stress resistance and then promote cell division (Confirmed in the fifth paragraph of Discussion, the increase of stress hormones/growth-promoting hormones ratio indicated an enhancement of plant stress resistance, the reduction of stress hormones/growth-promoting hormones ratio indicated that cell division was promoted)

 The fifth sentence (Qp_Lss was the most suitable indicator for evaluating wound conditions) was confirmed by principle component analysis in the sixth paragraph of discussion.

Response to Reviewer 2 Comments

Dear Professor

Thank you very much for your help. We are grateful for the constructive comments. We have addressed all the comments. Below is a detailed response to your comments.

Comments 1: Authors described photosynthesis was measured between 9 and 11 am, but all results indicated 4, 12 and 36 hrs after wounding. A reviewer can not understand how this time point was possible to measure all physiological parameters. For instance, if 4 hrs is morning, 12 and 36 hrs are night. In this experimental condition, hormone levels as well as photosynthesis and fluorescence can entirely affected by a diurnal fluctuation. How does authors maintain CO2 concentration and light intensity?

Response 1: Thank you very much for helping us detect the experimental process which was not clearly described in our manuscript. The experimental process is as follows: Before experiment, the seedlings were cultivated in growth chamber (200 μmol m–2 s–1 photosynthetic photon flux density under a 12-hour photoperiod from 8:00 am to 20:00 pm). At 5:00 am, the compression treatments were performed and seedlings were wounded. At 9:00 am (4 hrs after wounding), 17:00 pm (12 hrs after wounding), and 17:00 pm (next day, 36 hrs after wounding), photosynthesis, fluorescence and other parameters were measured. During photosynthesis measurement, CO2 concentration and light intensity in leaf chamber of portable photosynthesis system were separately set to 390 cm3 m-3 and 1000 μmol photon m-2 s-1. The experimental process was not described clearly due to our negligence, we revised it in the revised manuscript. Thank you again!  

Comments 2: A reviewer does not agree with authors argument. Show an evidence of stomatal closure and cell division. What does ‘stress resistance’ mean? This description of conclusion makes a big jump in logic assessment.

Response 2: Thank you very much. The conclusion is ambiguous due to improper description. The conclusion was revised in the manuscript.

In the revised manuscript, the conclusion consists of five sentences.

The first and second sentences (In conclusion, ……at 12 hours after wounding) described the impact of mechanical wounding on photosynthesis and fluorescence. At 4 hours after wounding, stomatal closure became the prominent factor limiting photosynthesis. However, at 12 hours after wounding, the non-stomatal factor became the prominent factor limiting photosynthesis (Confirmed in the first paragraph of Discussion). At 4 hours after wounding, the photoprotective mechanism prevented photodamage. However, at 12 hours after wounding, photodamage occurred (Confirmed in the second paragraph of Discussion).

The third and forth sentences (During mechanical wounding, various endogenous……promote cell division, respectively.) described the changes of hormones and hormonal interactions after mechanical wounding. Among them, various endogenous hormones were increased, which would accelerate wound healing (Confirmed in the fourth paragraph of Discussion). And also, the ratio of stress to growth-promoting hormones initially increased and then decreased, which initially enhance stress resistance and then promote cell division (Confirmed in the fifth paragraph of Discussion, the increase of stress hormones/growth-promoting hormones ratio indicated an enhancement of plant stress resistance, the reduction of stress hormones/growth-promoting hormones ratio indicated that cell division was promoted)

 The fifth sentence (Qp_Lss was the most suitable indicator for evaluating wound conditions) was confirmed by principle component analysis in the sixth paragraph of discussion.

Response to Reviewer 2 Comments

Dear Professor

Thank you very much for your help. We are grateful for the constructive comments. We have addressed all the comments. Below is a detailed response to your comments.

Comments 1: Authors described photosynthesis was measured between 9 and 11 am, but all results indicated 4, 12 and 36 hrs after wounding. A reviewer can not understand how this time point was possible to measure all physiological parameters. For instance, if 4 hrs is morning, 12 and 36 hrs are night. In this experimental condition, hormone levels as well as photosynthesis and fluorescence can entirely affected by a diurnal fluctuation. How does authors maintain CO2 concentration and light intensity?

Response 1: Thank you very much for helping us detect the experimental process which was not clearly described in our manuscript. The experimental process is as follows: Before experiment, the seedlings were cultivated in growth chamber (200 μmol m–2 s–1 photosynthetic photon flux density under a 12-hour photoperiod from 8:00 am to 20:00 pm). At 5:00 am, the compression treatments were performed and seedlings were wounded. At 9:00 am (4 hrs after wounding), 17:00 pm (12 hrs after wounding), and 17:00 pm (next day, 36 hrs after wounding), photosynthesis, fluorescence and other parameters were measured. During photosynthesis measurement, CO2 concentration and light intensity in leaf chamber of portable photosynthesis system were separately set to 390 cm3 m-3 and 1000 μmol photon m-2 s-1. The experimental process was not described clearly due to our negligence, we revised it in the revised manuscript. Thank you again!  

Comments 2: A reviewer does not agree with authors argument. Show an evidence of stomatal closure and cell division. What does ‘stress resistance’ mean? This description of conclusion makes a big jump in logic assessment.

Response 2: Thank you very much. The conclusion is ambiguous due to improper description. The conclusion was revised in the manuscript.

In the revised manuscript, the conclusion consists of five sentences.

The first and second sentences (In conclusion, ……at 12 hours after wounding) described the impact of mechanical wounding on photosynthesis and fluorescence. At 4 hours after wounding, stomatal closure became the prominent factor limiting photosynthesis. However, at 12 hours after wounding, the non-stomatal factor became the prominent factor limiting photosynthesis (Confirmed in the first paragraph of Discussion). At 4 hours after wounding, the photoprotective mechanism prevented photodamage. However, at 12 hours after wounding, photodamage occurred (Confirmed in the second paragraph of Discussion).

The third and forth sentences (During mechanical wounding, various endogenous……promote cell division, respectively.) described the changes of hormones and hormonal interactions after mechanical wounding. Among them, various endogenous hormones were increased, which would accelerate wound healing (Confirmed in the fourth paragraph of Discussion). And also, the ratio of stress to growth-promoting hormones initially increased and then decreased, which initially enhance stress resistance and then promote cell division (Confirmed in the fifth paragraph of Discussion, the increase of stress hormones/growth-promoting hormones ratio indicated an enhancement of plant stress resistance, the reduction of stress hormones/growth-promoting hormones ratio indicated that cell division was promoted)

 The fifth sentence (Qp_Lss was the most suitable indicator for evaluating wound conditions) was confirmed by principle component analysis in the sixth paragraph of discussion.

Response to Reviewer 2 Comments

Dear Professor

Thank you very much for your help. We are grateful for the constructive comments. We have addressed all the comments. Below is a detailed response to your comments.

Comments 1: Authors described photosynthesis was measured between 9 and 11 am, but all results indicated 4, 12 and 36 hrs after wounding. A reviewer can not understand how this time point was possible to measure all physiological parameters. For instance, if 4 hrs is morning, 12 and 36 hrs are night. In this experimental condition, hormone levels as well as photosynthesis and fluorescence can entirely affected by a diurnal fluctuation. How does authors maintain CO2 concentration and light intensity?

Response 1: Thank you very much for helping us detect the experimental process which was not clearly described in our manuscript. The experimental process is as follows: Before experiment, the seedlings were cultivated in growth chamber (200 μmol m–2 s–1 photosynthetic photon flux density under a 12-hour photoperiod from 8:00 am to 20:00 pm). At 5:00 am, the compression treatments were performed and seedlings were wounded. At 9:00 am (4 hrs after wounding), 17:00 pm (12 hrs after wounding), and 17:00 pm (next day, 36 hrs after wounding), photosynthesis, fluorescence and other parameters were measured. During photosynthesis measurement, CO2 concentration and light intensity in leaf chamber of portable photosynthesis system were separately set to 390 cm3 m-3 and 1000 μmol photon m-2 s-1. The experimental process was not described clearly due to our negligence, we revised it in the revised manuscript. Thank you again!  

Comments 2: A reviewer does not agree with authors argument. Show an evidence of stomatal closure and cell division. What does ‘stress resistance’ mean? This description of conclusion makes a big jump in logic assessment.

Response 2: Thank you very much. The conclusion is ambiguous due to improper description. The conclusion was revised in the manuscript.

In the revised manuscript, the conclusion consists of five sentences.

The first and second sentences (In conclusion, ……at 12 hours after wounding) described the impact of mechanical wounding on photosynthesis and fluorescence. At 4 hours after wounding, stomatal closure became the prominent factor limiting photosynthesis. However, at 12 hours after wounding, the non-stomatal factor became the prominent factor limiting photosynthesis (Confirmed in the first paragraph of Discussion). At 4 hours after wounding, the photoprotective mechanism prevented photodamage. However, at 12 hours after wounding, photodamage occurred (Confirmed in the second paragraph of Discussion).

The third and forth sentences (During mechanical wounding, various endogenous……promote cell division, respectively.) described the changes of hormones and hormonal interactions after mechanical wounding. Among them, various endogenous hormones were increased, which would accelerate wound healing (Confirmed in the fourth paragraph of Discussion). And also, the ratio of stress to growth-promoting hormones initially increased and then decreased, which initially enhance stress resistance and then promote cell division (Confirmed in the fifth paragraph of Discussion, the increase of stress hormones/growth-promoting hormones ratio indicated an enhancement of plant stress resistance, the reduction of stress hormones/growth-promoting hormones ratio indicated that cell division was promoted)

 The fifth sentence (Qp_Lss was the most suitable indicator for evaluating wound conditions) was confirmed by principle component analysis in the sixth paragraph of discussion.

Response to Reviewer 2 Comments

Dear Professor

Thank you very much for your help. We are grateful for the constructive comments. We have addressed all the comments. Below is a detailed response to your comments.

Comments 1: Authors described photosynthesis was measured between 9 and 11 am, but all results indicated 4, 12 and 36 hrs after wounding. A reviewer can not understand how this time point was possible to measure all physiological parameters. For instance, if 4 hrs is morning, 12 and 36 hrs are night. In this experimental condition, hormone levels as well as photosynthesis and fluorescence can entirely affected by a diurnal fluctuation. How does authors maintain CO2 concentration and light intensity?

Response 1: Thank you very much for helping us detect the experimental process which was not clearly described in our manuscript. The experimental process is as follows: Before experiment, the seedlings were cultivated in growth chamber (200 μmol m–2 s–1 photosynthetic photon flux density under a 12-hour photoperiod from 8:00 am to 20:00 pm). At 5:00 am, the compression treatments were performed and seedlings were wounded. At 9:00 am (4 hrs after wounding), 17:00 pm (12 hrs after wounding), and 17:00 pm (next day, 36 hrs after wounding), photosynthesis, fluorescence and other parameters were measured. During photosynthesis measurement, CO2 concentration and light intensity in leaf chamber of portable photosynthesis system were separately set to 390 cm3 m-3 and 1000 μmol photon m-2 s-1. The experimental process was not described clearly due to our negligence, we revised it in the revised manuscript. Thank you again!  

Comments 2: A reviewer does not agree with authors argument. Show an evidence of stomatal closure and cell division. What does ‘stress resistance’ mean? This description of conclusion makes a big jump in logic assessment.

Response 2: Thank you very much. The conclusion is ambiguous due to improper description. The conclusion was revised in the manuscript.

In the revised manuscript, the conclusion consists of five sentences.

The first and second sentences (In conclusion, ……at 12 hours after wounding) described the impact of mechanical wounding on photosynthesis and fluorescence. At 4 hours after wounding, stomatal closure became the prominent factor limiting photosynthesis. However, at 12 hours after wounding, the non-stomatal factor became the prominent factor limiting photosynthesis (Confirmed in the first paragraph of Discussion). At 4 hours after wounding, the photoprotective mechanism prevented photodamage. However, at 12 hours after wounding, photodamage occurred (Confirmed in the second paragraph of Discussion).

The third and forth sentences (During mechanical wounding, various endogenous……promote cell division, respectively.) described the changes of hormones and hormonal interactions after mechanical wounding. Among them, various endogenous hormones were increased, which would accelerate wound healing (Confirmed in the fourth paragraph of Discussion). And also, the ratio of stress to growth-promoting hormones initially increased and then decreased, which initially enhance stress resistance and then promote cell division (Confirmed in the fifth paragraph of Discussion, the increase of stress hormones/growth-promoting hormones ratio indicated an enhancement of plant stress resistance, the reduction of stress hormones/growth-promoting hormones ratio indicated that cell division was promoted)

 The fifth sentence (Qp_Lss was the most suitable indicator for evaluating wound conditions) was confirmed by principle component analysis in the sixth paragraph of discussion.

Response to Reviewer 2 Comments

Dear Professor

Thank you very much for your help. We are grateful for the constructive comments. We have addressed all the comments. Below is a detailed response to your comments.

Comments 1: Authors described photosynthesis was measured between 9 and 11 am, but all results indicated 4, 12 and 36 hrs after wounding. A reviewer can not understand how this time point was possible to measure all physiological parameters. For instance, if 4 hrs is morning, 12 and 36 hrs are night. In this experimental condition, hormone levels as well as photosynthesis and fluorescence can entirely affected by a diurnal fluctuation. How does authors maintain CO2 concentration and light intensity?

Response 1: Thank you very much for helping us detect the experimental process which was not clearly described in our manuscript. The experimental process is as follows: Before experiment, the seedlings were cultivated in growth chamber (200 μmol m–2 s–1 photosynthetic photon flux density under a 12-hour photoperiod from 8:00 am to 20:00 pm). At 5:00 am, the compression treatments were performed and seedlings were wounded. At 9:00 am (4 hrs after wounding), 17:00 pm (12 hrs after wounding), and 17:00 pm (next day, 36 hrs after wounding), photosynthesis, fluorescence and other parameters were measured. During photosynthesis measurement, CO2 concentration and light intensity in leaf chamber of portable photosynthesis system were separately set to 390 cm3 m-3 and 1000 μmol photon m-2 s-1. The experimental process was not described clearly due to our negligence, we revised it in the revised manuscript. Thank you again!  

Comments 2: A reviewer does not agree with authors argument. Show an evidence of stomatal closure and cell division. What does ‘stress resistance’ mean? This description of conclusion makes a big jump in logic assessment.

Response 2: Thank you very much. The conclusion is ambiguous due to improper description. The conclusion was revised in the manuscript.

In the revised manuscript, the conclusion consists of five sentences.

The first and second sentences (In conclusion, ……at 12 hours after wounding) described the impact of mechanical wounding on photosynthesis and fluorescence. At 4 hours after wounding, stomatal closure became the prominent factor limiting photosynthesis. However, at 12 hours after wounding, the non-stomatal factor became the prominent factor limiting photosynthesis (Confirmed in the first paragraph of Discussion). At 4 hours after wounding, the photoprotective mechanism prevented photodamage. However, at 12 hours after wounding, photodamage occurred (Confirmed in the second paragraph of Discussion).

The third and forth sentences (During mechanical wounding, various endogenous……promote cell division, respectively.) described the changes of hormones and hormonal interactions after mechanical wounding. Among them, various endogenous hormones were increased, which would accelerate wound healing (Confirmed in the fourth paragraph of Discussion). And also, the ratio of stress to growth-promoting hormones initially increased and then decreased, which initially enhance stress resistance and then promote cell division (Confirmed in the fifth paragraph of Discussion, the increase of stress hormones/growth-promoting hormones ratio indicated an enhancement of plant stress resistance, the reduction of stress hormones/growth-promoting hormones ratio indicated that cell division was promoted)

 The fifth sentence (Qp_Lss was the most suitable indicator for evaluating wound conditions) was confirmed by principle component analysis in the sixth paragraph of discussion.

Response to Reviewer 2 Comments

Dear Professor

Thank you very much for your help. We are grateful for the constructive comments. We have addressed all the comments. Below is a detailed response to your comments.

Comments 1: Authors described photosynthesis was measured between 9 and 11 am, but all results indicated 4, 12 and 36 hrs after wounding. A reviewer can not understand how this time point was possible to measure all physiological parameters. For instance, if 4 hrs is morning, 12 and 36 hrs are night. In this experimental condition, hormone levels as well as photosynthesis and fluorescence can entirely affected by a diurnal fluctuation. How does authors maintain CO2 concentration and light intensity?

Response 1: Thank you very much for helping us detect the experimental process which was not clearly described in our manuscript. The experimental process is as follows: Before experiment, the seedlings were cultivated in growth chamber (200 μmol m–2 s–1 photosynthetic photon flux density under a 12-hour photoperiod from 8:00 am to 20:00 pm). At 5:00 am, the compression treatments were performed and seedlings were wounded. At 9:00 am (4 hrs after wounding), 17:00 pm (12 hrs after wounding), and 17:00 pm (next day, 36 hrs after wounding), photosynthesis, fluorescence and other parameters were measured. During photosynthesis measurement, CO2 concentration and light intensity in leaf chamber of portable photosynthesis system were separately set to 390 cm3 m-3 and 1000 μmol photon m-2 s-1. The experimental process was not described clearly due to our negligence, we revised it in the revised manuscript. Thank you again!  

Comments 2: A reviewer does not agree with authors argument. Show an evidence of stomatal closure and cell division. What does ‘stress resistance’ mean? This description of conclusion makes a big jump in logic assessment.

Response 2: Thank you very much. The conclusion is ambiguous due to improper description. The conclusion was revised in the manuscript.

In the revised manuscript, the conclusion consists of five sentences.

The first and second sentences (In conclusion, ……at 12 hours after wounding) described the impact of mechanical wounding on photosynthesis and fluorescence. At 4 hours after wounding, stomatal closure became the prominent factor limiting photosynthesis. However, at 12 hours after wounding, the non-stomatal factor became the prominent factor limiting photosynthesis (Confirmed in the first paragraph of Discussion). At 4 hours after wounding, the photoprotective mechanism prevented photodamage. However, at 12 hours after wounding, photodamage occurred (Confirmed in the second paragraph of Discussion).

The third and forth sentences (During mechanical wounding, various endogenous……promote cell division, respectively.) described the changes of hormones and hormonal interactions after mechanical wounding. Among them, various endogenous hormones were increased, which would accelerate wound healing (Confirmed in the fourth paragraph of Discussion). And also, the ratio of stress to growth-promoting hormones initially increased and then decreased, which initially enhance stress resistance and then promote cell division (Confirmed in the fifth paragraph of Discussion, the increase of stress hormones/growth-promoting hormones ratio indicated an enhancement of plant stress resistance, the reduction of stress hormones/growth-promoting hormones ratio indicated that cell division was promoted)

 The fifth sentence (Qp_Lss was the most suitable indicator for evaluating wound conditions) was confirmed by principle component analysis in the sixth paragraph of discussion.

Reviewer 3 Report

Comments and Suggestions for Authors

This manuscript is generally well written. However, the authors need to address the following issues before accepting for publication in “Plants”.

1.     “Introduction” – the last short 4 paragraphs could be merged into 2 paragraphs. For example, Paragraph starting with “Hormonal signals may be involved in the regulation of several physiological processes……” could be merged with the paragraph “Machinal wounding…”. The last 2 paragraphs should also be merged into one.

2.     Section 2.2 Compression treatment: … “three different compression pressures (0, 40, and 80 N) were applied…” why used 40 and 80N?

3.     Section 2.3. Photosynthesis measurements: “Matured leaves …”, which leaves are matured leaves”?

4.     Section 2.3. Photosynthesis measurements: “The photosynthetic rate (Pn), stomatal conductance (gs) …” should be replaced by Asat and gs sat, respectively if “light intensity of 1000 μmol photon m-2 s-1.” is a saturated light intensity.

5.     For the three compress treatments, “0 N” instead of “control”, should be used throughout the paper including all figures.  Since “Tukey’s test” was used for data analysis, the comparisons are made among the three different compression pressures not between the control and each of the  treatment.

6.     In the discussion section, authors should draw some connections among the different parameters in response to mechanical wounding.

Author Response

Response to Reviewer 3 Comments

Dear Professor

Thank you very much for your help. We are grateful for the constructive comments. We have addressed all the comments. Below is a detailed response to your comments.

Comments 1: “Introduction” – the last short 4 paragraphs could be merged into 2 paragraphs. For example, Paragraph starting with “Hormonal signals may be involved in the regulation of several physiological processes……” could be merged with the paragraph “Machinal wounding…”. The last 2 paragraphs should also be merged into one.

Response 1: Thank you very much. The last short 4 paragraphs have been merged into 2 paragraphs in the revised manuscript.  

Comments 2: Section 2.2 Compression treatment: … “three different compression pressures (0, 40, and 80 N) were applied…” why used 40 and 80N?

Response 2: Thank you very much. Affected by multiple factors, the clamping force exist very large difference during seedling transplantation. We found that irreversible deformation of seedling stems occurred under 40 N pressure, and 80 N pressure is the limit of clamping force.

Comments 3: Section 2.3. Photosynthesis measurements: “Matured leaves …”, which leaves are matured leaves”?

Response 3: Thank you very much. In our experiment, The second fully expanded functional leaves were selected for gas exchange measurements. The mistake was revised in the manuscript.

Comments 4: Section 2.3. Photosynthesis measurements: “The photosynthetic rate (Pn), stomatal conductance (gs) …” should be replaced by Asat and gssat, respectively if “light intensity of 1000 μmol photon m-2 s-1.” is a saturated light intensity.

Response 4: Thank you very much. In the experiment, the instantaneous values of photosynthetic rate (Pn), stomatal conductance (gs), ……, using a portable photosynthesis system with a CO2 concentration of 390 cm3 m-3 and a light intensity of 1000 μmol photon m-2 s-1. Because we did not measure the light response curves, we are afraid it is not enough to determine the instantaneous values are Asat and gssat. Thank you again for providing us with ideas for further research.

Comments 5: For the three compress treatments, “0 N” instead of “control”, should be used throughout the paper including all figures.  Since “Tukey’s test” was used for data analysis, the comparisons are made among the three different compression pressures not between the control and each of the  treatment.

Response 5: Thank you very much. The mistake was revised in the manuscript.

Comments 6: In the discussion section, authors should draw some connections among the different parameters in response to mechanical wounding.

Response 6: Thank you very much for your suggestion. We have considered analyzing the relationship between different parameters once. The main aim of this study was to screen the key parameters reflecting the mechanical wounding conditions, therefore a large amount of studies about principal component analysis (PCA) was performed, and the relationship between different parameters were not analyzed in this manuscript. We will analyze the relationship between different parameters in further study. Thank you again for providing us with ideas for further research.

Round 2

Reviewer 2 Report

Comments and Suggestions for Authors

Dear authors,

Author Response

Dear Professor

Thank you very much for your help. We are grateful for your constructive comments, which not only improved the quality of our manuscript, but also provided us innovative ideas for further research. We have addressed all the comments refer to the manuscript. Below is a detailed response to your comments.

Comments 1: Light period was 8 am to 20 pm and photosynthesis was measured at 9 am and 17 pm. Based on usual daily fluctuation of photosynthesis, two time points measuring photosynthesis is hard to be represented as real photosynthetic activity of plant due to indicating both time points, daily initiation and end, respectively. Authors should explain this point.

Response 1: Thank you very much. Exactly, the diurnal variation of plant photosynthesis resulted in the differences in photosynthetic measurement at different time points. In the study, lots of work have been done to minimize the differences. To avoid the impact of diurnal variation on plant photosynthesis, photosynthetic measurements were carried out in growth chamber which provide homogeneous conditions of light, temperature and humidity from 8:00 am to 20:00 pm. More importantly, at each time point, photosynthetic measurements for both control and wounded seedlings were carried out. The impact of mechanical wounding on seedlings could be evaluated by analyzing the photosynthetic difference between control and wounded seedlings at each time point.

Comments 2: Additionally, a reviewer is wondering whether the strength of wounding stress affected xylem and phloem of stem. Please show microscopic observation or photo to support the effect of endogenous hormones such as IAA, ZR and GA to recover stem tissue from wounding stress from 0, 4, 12 and 36 hrs after treatment if authors have the results.

Response 2: Thank you very much for your suggestion. According to our research progress in this field, the microscopic experiment has not yet been conducted. Your suggestion points out the way to us for further research. We will carried out the researches in this field further. Thank you very much again!

Comments 3: A review recommends to combine Fig. 5 and 6. It is better for reader’s understanding.

Response 3: Thank you very much. Fig. 5 and Fig. 6 were combined in the revised manuscript.

Comments 4: In conclusion, what is non-stomatal factor? Please describe in more detail.

Response 4: Thank you very much. Except stomatal closure, lots of non-stomatal factors would limit photosynthesis. In our study, photosynthetic non-stomatal limitation caused by photodamage became the prominent factor limiting photosynthesis. The details were described in the revised manuscript.

Thank you very much again for your constructive comments, which not only improved the quality of our manuscript, but also provided us innovative ideas for further research.

Sincerely,

Xin Jin

College of Agricultural Equipment Engineering, Henan University of Science and Technology, Luoyang 471000, China

E-mail address: jx.771@163.com

Reviewer 3 Report

Comments and Suggestions for Authors

The authors have addressed most of my comments.

Author Response

Dear Professor

Thank you very much for your help. Below is the response to your comments.

Comments and suggestions for Authors: The authors have addressed most of my comments.

Response: Thank you very much for your help. We are grateful for your constructive comments, which not only improved the quality of our manuscript, but also provided us innovative ideas for further research.  Thank you very much again !

Sincerely,

Xin Jin

College of Agricultural Equipment Engineering, Henan University of Science and Technology, Luoyang 471000, China

E-mail address: jx.771@163.com